# Binocular Visual Measurement Method Based on Feature Matching

**DOI:** 10.3390/s24061807

**Published:** 2024-03-11

**Authors:** Zhongyang Xie, Chengyu Yang

**Affiliations:** School of Electro-Optical Engineering, Changchun University of Science and Technology, Changchun 130022, China; 2021100355@mails.cust.edu.cn

**Keywords:** binocular vision, three-dimensional measurement, image processing, feature matching

## Abstract

To address the issues of low measurement accuracy and unstable results when using binocular cameras to detect objects with sparse surface textures, weak surface textures, occluded surfaces, low-contrast surfaces, and surfaces with intense lighting variations, a three-dimensional measurement method based on an improved feature matching algorithm is proposed. Initially, features are extracted from the left and right images obtained by the binocular camera. The extracted feature points serve as seed points, and a one-dimensional search space is established accurately based on the disparity continuity and epipolar constraints. The optimal search range and seed point quantity are obtained using the particle swarm optimization algorithm. The zero-mean normalized cross-correlation coefficient is employed as a similarity measure function for region growing. Subsequently, the left and right images are matched based on the grayscale information of the feature regions, and seed point matching is performed within each matching region. Finally, the obtained matching pairs are used to calculate the three-dimensional information of the target object using the triangulation formula. The proposed algorithm significantly enhances matching accuracy while reducing algorithm complexity. Experimental results on the Middlebury dataset show an average relative error of 0.75% and an average measurement time of 0.82 s. The error matching rate of the proposed image matching algorithm is 2.02%, and the PSNR is 34 dB. The algorithm improves the measurement accuracy for objects with sparse or weak textures, demonstrating robustness against brightness variations and noise interference.

## 1. Introduction

Binocular vision, as a primary implementation of three-dimensional visual measurement, has advantages such as non-contact measurement, low cost, and simple construction. It can detect information about the flatness, relative height, size, and depth of objects. It is widely applied in areas like industrial automation, agricultural production, virtual reality, and unmanned driving [1]. Improving the accuracy and robustness of binocular vision measurement systems can provide support for high-precision fields such as robotic navigation and medical imaging, and addresses the complexities of measurement in various environments. The principle of binocular vision is based on the parallax principle, utilizing imaging devices to capture two images of the measured object from different positions. By calculating the positional deviation between corresponding points in the images, three-dimensional geometric information is obtained. Image feature matching is a crucial step for binocular cameras to acquire depth information. Improving the feature matching algorithm can reduce matching errors, handle various complex scenes, and enhance the performance of binocular cameras. The main challenge lies in whether same-named pixels in the left and right images can be matched quickly and accurately [2].

To enhance feature matching between images, numerous scholars have researched it from different perspectives [3,4,5,6,7,8,9,10,11,12]. Based on the matching method, it can be categorized into deep-learning-based matching algorithms and artificial-feature-based matching algorithms. Among them, some of the more classical ones include Bromley et al.’s [13] proposed method for matching homonymous pixels by establishing a Siamese network. This network structure comprises two subnetworks with shared weights. These subnetworks, located at the front end of the overall network, aim to process input signals in the same way. In the Siamese network used for binocular stereo matching, these two weight-sharing subnetworks handle left and right images separately. The features obtained through convolution are transformed into feature vectors by fully connected layers. The similarity between two images is determined by calculating the Manhattan distance between the two feature vectors. Siamese networks involve larger computational complexity and longer training times compared with conventional networks due to their dual inputs and subnetworks. Moreover, their output is the distance between two classes rather than a probability, making them less practical. Pilzer et al. [14] introduced a generative adversarial network (GAN) model for binocular stereo matching, consisting of two symmetric generator models and one discriminator model. The symmetric generator models generate images from different perspectives, and their outputs are fused through mutual constraints and supervision to produce the final disparity map. However, GANs often yield unstable results, may not satisfy the desired distribution, require expertise and skills during training, and involve complex structures and optimization algorithms, consuming substantial time and computational resources. Wu Weiren et al. [15] applied graph cuts to solve the stereo matching problem, asserting that graph cut algorithms can address the issue of discontinuous disparities. While graph cut algorithms offer high accuracy and fast convergence, their drawback lies in long computation times and poor real-time performance. Wang et al. [16] treated regions as the matching primitives, minimizing matching costs through cooperative optimization between regions. The algorithm is region-based, featuring simplicity in computation and fast processing. However, it is susceptible to misalignment in weak-textured areas, is influenced by lighting and occlusion, and poses challenges in window selection. Zhang Jieyu et al. [17] explored image registration technology using the SIFT feature extraction algorithm. This method demonstrates robustness to lighting changes and noise but involves complex preprocessing. The accuracy of feature matching algorithms is directly influenced by feature extraction and the search strategy for similar points. The sparse nature of extracted matching features makes it challenging to generate dense disparity maps.

This paper improves the matching of image feature points and adopts a method combining feature points with feature regions to enhance the measurement accuracy and performance of binocular cameras. Initially, the features from accelerated segment test (FAST) [18] is utilized to detect and extract feature information from the left and right images. The obtained feature points are then grown based on the continuity criterion of disparity, and the matching cost is computed using the zero-mean normalized cross-correlation (ZNCC) as a similarity measure. The minimum cost point pairs are identified, and some mismatches are eliminated through thresholding. The complexity of the search is reduced by employing epipolar line constraints. The particle swarm optimization algorithm is employed to determine the optimal search range and the number of seed points. Subsequently, matching is performed between the left and right images based on the grayscale information of feature regions. Seed point matching is further conducted within each matching region. Finally, the obtained matching pairs are used to calculate the three-dimensional information of the object under measurement using the triangulation formula. Compared with existing methods, the algorithm proposed in this paper is based on region growing, combining the advantages of feature point image matching and feature region matching. This significantly improves the accuracy of matching while reducing the complexity of the algorithm. Experimental results on the Middlebury dataset demonstrate that the algorithm proposed in this paper improves the measurement accuracy of binocular cameras for objects with sparse texture and weak texture, while also demonstrating strong robustness, capable of suppressing interference from certain brightness differences and noise.

## 2. Algorithm Flow Design

The fundamental theoretical logic of stereoscopic vision sensors for 3D measurement is as follows: Two cameras capture the same scene at different viewpoints, and by performing stereo matching, corresponding pixels are identified. The horizontal difference in the coordinates of these corresponding pixels represents the corresponding disparity information. Subsequently, three-dimensional information is calculated based on the principles of triangulation [19].

The specific steps are as follows:(1)Obtain left and right images from the stereo camera.(2)Utilize the FAST feature point detection algorithm for feature extraction.(3)Treat feature points as seed points and perform region growing.(4)Based on the information from feature regions, match the left and right images.(5)Within each pair of matched feature regions, match the seed points.(6)If the majority of feature points in a region do not match, it indicates a region matching error, and the calculation is returned. If the majority of feature points within a region are correct, the region matching is considered correct, and incorrect feature point matches are eliminated.(7)Use the triangulation formula to calculate the three-dimensional information of the object under test.

The algorithm flow is illustrated in Figure 1 below.

## 3. Three-Dimensional Measurement Method Based on Improved Feature Matching Algorithm

### 3.1. FAST Feature Point Detection

Feature points in an image refer to those points that are prominent, do not disappear due to changes in lighting conditions, and have rotation invariances, such as corner points, edge points, bright points in dark areas, and dark points in bright areas. The FAST algorithm employs an efficient pixel comparison strategy, demonstrating excellent performance in real-time applications and resource-constrained environments.

The basic idea of FAST feature point detection is to consider a pixel as the center and, when the characteristic differences between other pixels on a circle with a certain radius and the central pixel meet a standard, recognize that point as a corner point.

### 3.2. Region Growth Based on Seed Points

In this paper, the obtained feature points are used as seed points. Based on the continuity of disparity and accurate epipolar line constraints, a one-dimensional search space is established. The particle swarm optimization algorithm is employed to obtain the optimal search range and the number of seed points. The zero-mean normalized cross-correlation coefficient is used as the similarity measure function to achieve region growth.

#### 3.2.1. Constraints of Binocular Cameras

Due to the structural characteristics of binocular cameras, the left and right images exhibit certain regularities in information content. These regularities are referred to as constraints of binocular cameras, with the most common ones being the epipolar line constraint and the disparity gradient constraint.

The epipolar line constraint of a stereo camera describes the constraints formed by the projection of the same point onto two different views in the projection model, involving the points and camera optical centers. As shown in Figure 2, P(x,y,z) is the point to be tested; $P_{l}$ and $P_{r}$ are the mapped points in the left and right images, respectively; $O_{l}O_{r}$ is the baseline between the two cameras; the intersection points of the baseline with the left and right views are denoted as $e_{l}$ and $e_{r}$; plane $O_{l}O_{r}P$ is referred to as the epipolar plane; the intersection lines $e_{l}p_{l}$ and $e_{r}p_{r}$ between the epipolar plane and the left and right image planes are called epipolar lines.

The epipolar line constraint implies that when the point
P(x,y,z) to be tested is imaged in both the left and right images, given the known coordinates of the 
mapped point $P_{l}$ in the left image, the coordinates of the mapped point $P_{r}$ in the right image can be found on the epipolar plane $O_{l}O_{r}P$.

The disparity gradient constraint of a binocular camera refers to the process of calculating gradients on the disparity image, using the rate of change of disparity values to constrain the depth estimation. The disparity gradient constraint helps eliminate discontinuities in the depth image, enhancing the accuracy of depth estimation. It can also be applied to the depth image by computing gradients, reducing depth estimation errors caused by matching mistakes. As shown in Figure 3, m and n are two adjacent corners in the left image, $m^{\prime }$ and $n^{\prime }$ are two adjacent corners in the right image, $\left(m,m^{\prime }\right)$ represents one pair of matched points, and $\left(n,n^{\prime }\right)$ is another pair of matched points. The equation for calculating the disparity gradient $G_{d}$ is as follows:(1)$$G_{d}=2\frac{\left\|\left(m^{\prime }-m\right)-\left(n^{\prime }-n\right)\right\|}{\left\|\left(m^{\prime }-m\right)+\left(n^{\prime }-n\right)\right\|}$$

If points $\left(m,m^{\prime }\right)$ and $\left(n,n^{\prime }\right)$ from two matched pairs are mutually matched, then $G_{d}$ should be less than or equal to 2. If the value of $G_{d}$ is greater than 2, it can be considered that the matching of the feature points is incorrect.

#### 3.2.2. Improved Particle Swarm Optimization

The particle swarm optimization algorithm is a simple and effective stochastic global optimization technique inspired by the behavior of flocks of birds, aiming to find the optimal solution through collaboration and information sharing among individuals. In the particle swarm algorithm, each solution to the optimization problem is represented as a bird in the search space, also referred to as a “particle”. Each particle has an adaptive value determined by the function being optimized, and a velocity that determines its direction and distance of flight. The particles then follow the current optimal particle in the solution space [20].

A challenging aspect of the proposed algorithm is dealing with the quantity of feature points and obtaining feature regions. In general, a large number of feature points appear in the image space, and each feature point needs to spread in conjunction with the disparity gradient to surrounding image pixels, which may affect the measurement speed. Conversely, if there are fewer feature points, many regions may not become feature regions as they are not reached by the spread of feature points, affecting the accuracy of matching. To improve the speed and accuracy of the measurement, the particle swarm optimization algorithm is employed for enhancement. The spreading process’s search range is used as the particle’s velocity, and the quantity of feature points is used as the particle’s position. By seeking the global optimum, the most suitable parameter values are obtained to enhance the algorithm’s speed and accuracy [21].

When dealing with high-dimensional complex problems, particle swarm algorithms often face challenges such as becoming stuck in local optimal solutions, slow convergence in the later stages of iteration, and low solution accuracy. To address the tendency of particle swarm optimization algorithms to converge to local minimum values, this paper proposes an improved particle swarm optimization algorithm. When particles are trapped in poor search areas, a stretching operation is performed on these trapped particles with a certain probability, pulling them from the poorer region towards the better region discovered so far. This helps the trapped particles escape from poor regions, search for better regions, and reasonably allocate search resources. This approach reduces the probability of particles converging to local optimal solutions to some extent, increasing the likelihood of particles finding global optimal solutions and potentially achieving higher precision solutions [22].

The specific improvement steps are as follows:
(1)Suppose in a D-dimensional target search space, there is a particle population $x=[x_{1},x_{2},x_{3},\cdots ,x_{M}]$ consisting of *M* particles representing potential solutions to the problem. Each particle has a position and velocity, where the position of the *i*-th particle is represented as a *D*-dimensional vector $x_{i}=[x_{i1},x_{i2},x_{i3},\cdots ,x_{iD}]$, and the velocity of the *i*-th particle is represented as a *D*-dimensional vector $x_{i}=[x_{i1},x_{i2},x_{i3},\cdots ,x_{iD}]$.(2)Start by randomly initializing *M* particles and then iteratively find the optimal solution. In each iteration, each particle updates its velocity and position based on two extremes to guide its flight. One extreme is the local optimal solution for that particle, and the other extreme is the global optimal solution. The local optimal solution is the best value found by each particle up to the current iteration, denoted as $pbest_{i}=[p_{i1},p_{i2},p_{i3},\cdots ,p_{iD}]$. The global optimal solution is the best value found by the entire particle swarm up to the current iteration, denoted as $gbest=[p_{g1},p_{g2},p_{g3},\cdots ,p_{gD}]$.(3)After updating the local and global optimal solutions, particles update their velocity and position according to the following Equations (2) and (3).
(2)$$\begin{array}{ll} v_{id}(i+1)= & w\times v_{id}(t)+c_{1}\times rand()\times (pbest_{id}-x_{id}(t))+\\ & c_{2}\times rand()\times (gbest_{d}-x_{id}(t)) \end{array}$$
(3)$$x_{id}(i+1)=x_{id}(t)+v_{id}(t+1)$$


In the equations, *t* represents the current iteration count; $c_{1}$ and $c_{2}$ are learning factors used to adjust the step size of particles flying towards individual and global extremes, respectively; w is called the inertia weight, typically taking values between 0.4 and 0.9.


(4)For each particle trapped in a poorer search region, define the stretching operation:(4)$$SO=c_{3}\times rand(\;)\times (gbest_{d}-pbest_{id})$$
(5)$$c_{3}=\left(f\left(i\right)-f_{\min}\right)/\left(f_{avg}-f_{\min}\right)$$


In the equation, $c_{3}$ is the adaptive stretching factor, f(i) is the fitness of particle *i*, $f_{\min}$ is the minimum fitness value of the particle swarm, and $f_{avg}$ is the average fitness of the particle swarm.

When the fitness of particle *i* is farther from the minimum fitness value of the particle swarm, the larger the value of the stretching factor $c_{3}$. Therefore, when this stretching operation takes effect, it has a greater impact on the particle’s flight speed, and consequently, a larger update to the particle’s position. Not all particles trapped in poorer search regions undergo stretching operations; this is only applied to particles whose fitness values have been greater than the population’s adaptive fitness for three consecutive evaluations. Implementing stretching operations for all particles trapped in poorer search regions reduces the diversity of the particle swarm, leading to premature convergence. In the study, this operation is probabilistically applied to allow particles to escape from the poorer search regions, increasing the likelihood of the particle swarm algorithm finding the global optimum or moving toward a higher-precision domain, thereby achieving a more balanced resource allocation.

#### 3.2.3. Similarity Measurement Function

This study utilizes the zero-mean normalized cross-correlation coefficient as a similarity measurement function to implement region growing. A window is set around the target pixel to calculate the matching cost, and a threshold is set to eliminate some mismatched points, reducing the influence of noise points. If the minimum matching cost meets the threshold, the corresponding matching point pair is recorded. The equation for calculating the zero-mean normalized cross-correlation coefficient is as follows:(6)$$ZNCC=\frac{\sum_{i\in W}\left(I_{L}\left(x_{L}+i\right)-{\bar{I}}_{L}\left(x_{L}\right)\right)\left(I_{R}\left(x_{R}+i\right)-{\bar{I}}_{R}\left(x_{R}\right)\right)}{\sqrt{\sum_{i\in W}{\left(I_{L}\left(x_{L}+i\right)-{\bar{I}}_{L}(x_{L})\right)}^{2}\sum_{i\in W}{\left(I_{R}\left(x_{R}+i\right)-{\bar{I}}_{R}\left(x_{R}\right)\right)}^{2}}}$$

In the equation, *W* represents the search window size, ${\bar{I}}_{L}(x_{L})$ and ${\bar{I}}_{R}\left(x_{R}\right)$ denote the average gray values of the search window *W* centered at $x_{L}$ and $x_{R}$ in the left and right images, respectively. $I_{L}(x_{L}+i)$ and $I_{R}(x_{R}+i)$ represent the gray values of points within the search window *W* centered at $x_{L}$ and $x_{R}$ in the left and right images, respectively.

#### 3.2.4. Region Growing

The process of region growing involves gradually aggregating a pixel or subregion into a complete and independent connected region according to a defined diffusion criterion. Let S represent a subset of pixels in an image. If there exists a path between all pixels in S, then two pixels P and Q in S are considered connected. Connectivity between pixels is a crucial concept in determining regions. In a two-dimensional image, assuming the target pixel has m adjacent pixels (m ≤ 8), if the grayscale value of this pixel is equal to the grayscale value of one of these m pixels, it is considered connected to pixel A. Common connectivities include 4-connectivity and 8-connectivity. For 4-connectivity, the target pixel’s four adjacent pixels are chosen (up, down, left, right). For 8-connectivity, all adjacent pixels in two-dimensional space are selected, including the target pixel’s top-left, top, top-right, right, bottom-right, bottom, bottom-left, and left neighbors.

The region-growing process is iterative. Each seed pixel undergoes iteration for growth. For the region of interest R in the image, Z represents seed points pre-discovered on region R. According to set criteria, pixels within a certain neighborhood of seed point Z that satisfy the similarity criterion are progressively merged into a seed group for the next stage of expansion. This iterative diffusion continues until the growth stop condition is met. This completes the process of expanding from a seed point to an independent connected region of interest. The region-growing process is illustrated in Figure 4 below.

In Figure 4, the red squares represent seed points, and the yellow squares represent points that become feature regions after diffusion.

### 3.3. Triangulation Principle

Binocular vision measurement technology is biologically inspired by the binocular vision of humans or primates. Human eyes are located at the front of the head, and there is an overlapping region in the field of view between the left and right eyes, known as the binocular overlapping area. Objects within this overlapping region exhibit relative disparity. After the visual information from the two eyes is processed by the brain’s higher central nervous system, the disparity information of objects can be converted into depth information. This allows humans to perceive the distance of objects in space and experience a three-dimensional stereoscopic visual effect.

Binocular cameras capture left and right images through CCD photosensitive devices. Due to the baseline distance between the two cameras, the two images exhibit disparity. The principle of binocular vision is based on the disparity images. Using the positional relationships derived from triangle similarity, the depth distance of the measured point P from the camera is calculated. The schematic diagram of the measurement process with a binocular camera is illustrated in Figure 5 below.

As shown in Figure 5, the distance between the two optical lenses is the baseline distance B of the measurement system, which is a fixed value. The image points of the measured point P(x,y,z) in the two cameras are $P_{r}$ and $P_{l}$, and the difference in their respective image coordinates is $\left|x_{r}-x_{l}\right|$, which represents the disparity distance. From the geometric optical position relationship, it can be seen that $\Delta PP_{r}P_{l}∼\Delta PO_{r}O_{l}$, so Equations (7) and (8) can be derived.
(7)$$\frac{x_{r}-x_{l}}{B}=\frac{z}{z+f}$$
(8)$$z=\frac{(x_{r}-x_{l})\times f}{\left(x_{r}-x_{l}-B\right)}$$

Similarly, coordinate information for point *P* can be obtained on the y-axis and z-axis.

## 4. Experimental Results and Data Analysis

### 4.1. Experimental Setup

This experiment was conducted on a computer platform based on the Windows 11 operating system and the Intel Core i9-11900 K processor. The software development environment for this experiment primarily included Visual Studio 2019, OpenCV 3.0, and Matlab R2020b. The experiment utilized images from the stereo datasets of the Middlebury test platform in 2006, specifically the bowling, lampshade, plastic, and wood image sets. These datasets were captured by Brad Hiebert-Treuer, Sarri Al Nashashibi, and Daniel Scharstein during the summer of 2006 at Middlebury College and were published in the CVPR 2007 conference. The focal length is 3740 pixels, and the baseline is 160 mm. The experiments for three-dimensional measurements were primarily conducted under normal lighting conditions of 30 cd/m^2^ brightness and in dry air, without being affected by other more stimulating environmental interferences. The selected image resolution is half size, with a width of 620 and a height of 555. The experiment involved subjective and objective comparisons of the proposed improvement algorithm with the use of the Siamese network algorithm, generative adversarial network (GAN) algorithm, region matching algorithm, DP algorithm [23], and SIFT matching algorithm.

### 4.2. Experimental Results

The results of the experiment are shown in the figure below. From the figure, it can be observed that the matching effects of the other five algorithms are poor at the edges, with many instances of noise and holes. The Siamese network algorithm and GAN algorithm, which employ neural networks, exhibit slight errors in the disparity information at the edges of objects. The disparity map processed by the DP algorithm appears overall blurry. The disparity map after applying the SIFT algorithm shows some clear boundaries but is noticeably inferior to the improvement algorithm proposed in this paper. Figure 6 is the experimental results for bowling stereo image pair. Figure 7 is the experimental results for lampshade stereo image pair. Figure 8 is the experimental results for plastic stereo image pair. Figure 9 is the experimental results for wood stereo image pair.

### 4.3. Data Analysis

In the field of stereo matching in images, the main evaluation criteria are the mismatch rate and peak signal-to-noise ratio (PSNR). The mismatch rate is measured by calculating the proportion of incorrectly matched pixels to the total number of pixels, as shown in Equation (9). The PSNR value is usually expressed in decibels (dB), with a higher value indicating better image quality. Equations (10) and (11) represent its calculation.
(9)$$B=\frac{1}{M}\sum_{(x,y)\in A}\left[|R(x,y)-O(x,y)|>\delta \right]$$

In the equations, *M* represents the total number of pixels in the entire image, R(x,y) is the calculated disparity map function, O(x,y) is the true disparity map function, and δ is the allowable error.
(10)$$PSNR=10\times \log_{10}(\frac{MAX_{i}^{2}}{MSE})$$
(11)$$MSE=\frac{1}{M}\sum_{\left(x,y\right)}||R(x,y)-O(x,y)|{|}^{2}$$

In the equation, *MAX* represents the maximum possible pixel value in the image, typically 255, indicating the maximum pixel value for an 8-bit image. *MSE* is the mean square error, representing the average pixel value difference between the original image and the processed image. In the experiments, data results for four sets of experiments were compiled for six algorithms, including the Siamese network algorithm, generative adversarial network (GAN) algorithm, region matching element algorithm, DP algorithm, SIFT matching algorithm, and the proposed improved algorithm. The summarized results are shown in Table 1.

From the above image data, it can be observed that the algorithm proposed in this paper performs better when dealing with situations involving sparse disparity, weak textures, and insufficient disparity accuracy, demonstrating strong robustness. Additionally, the algorithm’s complexity has been significantly reduced through improvements. The enhanced feature matching method, which avoids the neural network approach, eliminates the need for extensive data for training. It also deviates from a purely manual feature extraction route, enhancing the rationality of the execution steps. 

Table 2 compares the disparity results obtained by the proposed algorithm with the original image matching rate after increasing the brightness of the image by 10 cd/m^2^, 20 cd/m^2^, 40 cd/m^2^, and 50 cd/m^2^, respectively; 30 cd/m^2^ represents the original light intensity in the experiment. From the experimental results, it can be observed that the accuracy rate with 30 cd/m^2^ and the other four groups remains around 97.57%, with little difference. The experimental results demonstrate that the algorithm proposed in this paper exhibits strong robustness in the face of changes in illumination.

The feature matching results obtained in this study, combined with the principles of triangulation, are utilized to calculate the three-dimensional dimensions of objects. The specific measurement results are presented in Table 3.

Through comparing the detection results with the actual lengths, the relative measurement error is obtained, and the corresponding detection time is recorded after each experiment. From the results in Table 3, it can be observed that the relative error is generally around 1%, and the detection speed of three-dimensional information is mostly below 1 s, meeting the expected requirements.

## 5. Conclusions

This study focuses on improving the measurement of three-dimensional information from stereo cameras, addressing issues such as low measurement accuracy and unstable results when detecting objects with sparse surface textures, weak surface textures, occlusions, low contrast, and strong lighting variations. The proposed method is based on an enhanced feature matching triangulation approach. Initially, features are extracted from the left and right images obtained by the stereo camera, and these feature points are used as seed points. By establishing a one-dimensional search space based on the accuracy of disparity continuity and epipolar line constraints, the particle swarm optimization algorithm is employed to determine the optimal search range and the number of seed points. The zero-mean normalized cross-correlation coefficient is used as a similarity measure to achieve region growing. Subsequently, matching is performed based on the grayscale information of the feature regions in the left and right images, and seed point matching is carried out within each matching region. Finally, the obtained matching pairs are used with the triangulation formula to calculate the three-dimensional information of the object under test. The improved algorithm not only enhances the measurement accuracy for uniformly planar regions but also reduces computational complexity. The experiments were conducted using the bowling, lampshade, plastic, and wood image sets from the 2006 stereo datasets of the Middlebury official testing platform. Six algorithms, including the Siamese network algorithm, the generative adversarial network algorithm, the region matching element algorithm, the DP algorithm, the SIFT matching algorithm, and the proposed improved algorithm, were experimentally compared subjectively and objectively. In the end, ten sets of experimental data were measured and calculated, showing an average relative error of 0.75% and an average measurement time of 0.82 s, meeting the expected requirements.

## Figures and Tables

**Figure 1 sensors-24-01807-f001:**
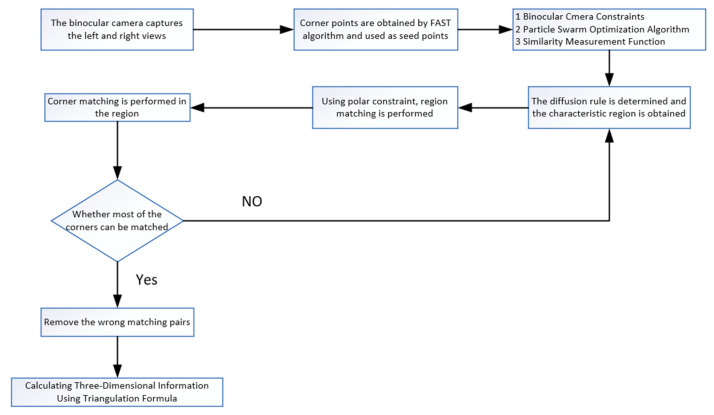
Three-dimensional measurement workflow.

**Figure 2 sensors-24-01807-f002:**
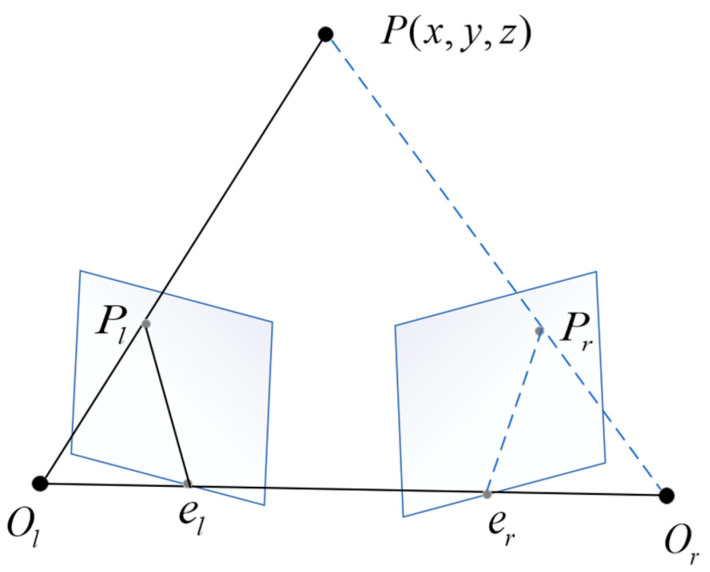
Epipolar line constraint.

**Figure 3 sensors-24-01807-f003:**
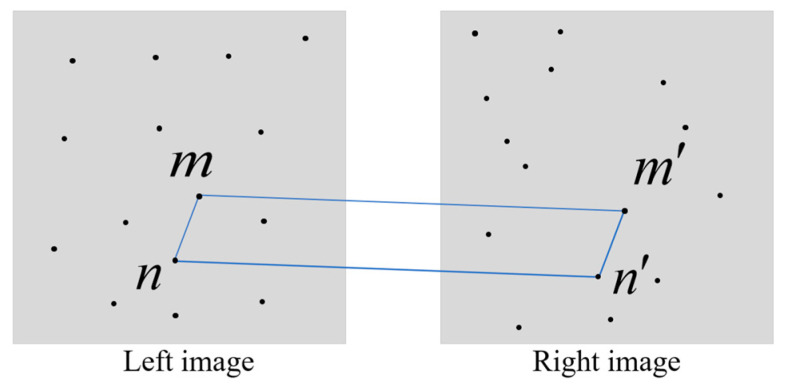
Disparity gradient constraint.

**Figure 4 sensors-24-01807-f004:**
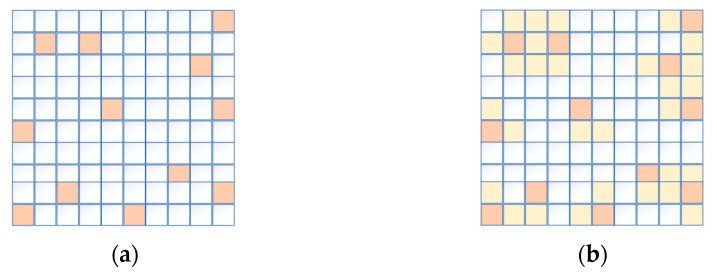
Region growing process. (**a**) Image of seed point pixel; (**b**) image after seed point diffusion.

**Figure 5 sensors-24-01807-f005:**
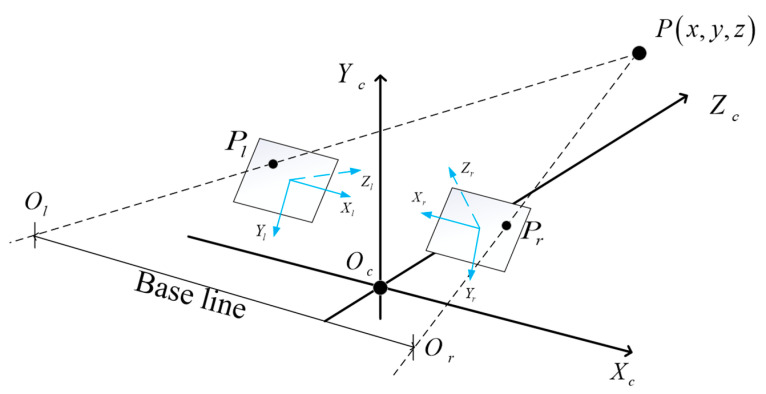
Schematic diagram of binocular camera measurement.

**Figure 6 sensors-24-01807-f006:**
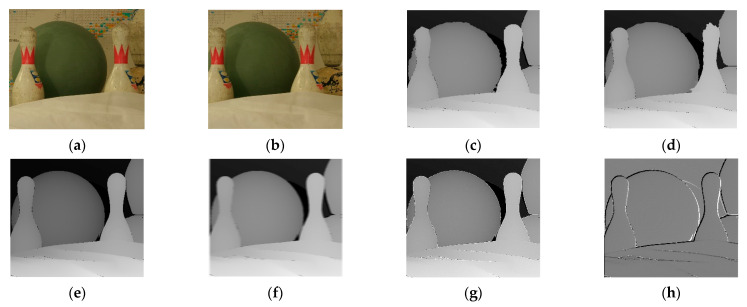
Experimental results for bowling stereo image pair. (**a**) Left view; (**b**) right view; (**c**) Siamese network algorithm; (**d**) generative adversarial network algorithm; (**e**) region matching algorithm; (**f**) DP algorithm; (**g**) SIFT matching algorithm; (**h**) proposed algorithm.

**Figure 7 sensors-24-01807-f007:**
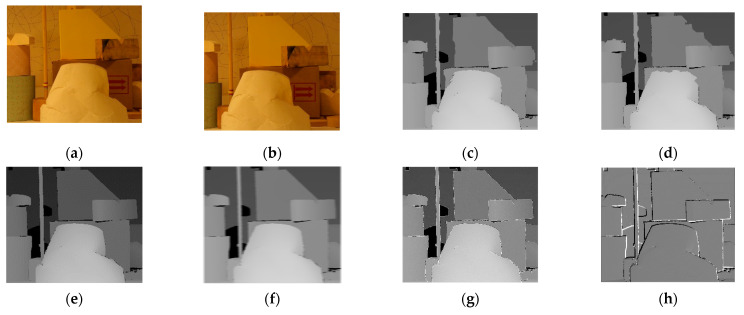
Experimental results for lampshade stereo image pair. (**a**) Left view; (**b**) right view; (**c**) Siamese network algorithm; (**d**) generative adversarial network algorithm; (**e**) region matching algorithm; (**f**) DP algorithm; (**g**) SIFT matching algorithm; (**h**) proposed algorithm.

**Figure 8 sensors-24-01807-f008:**
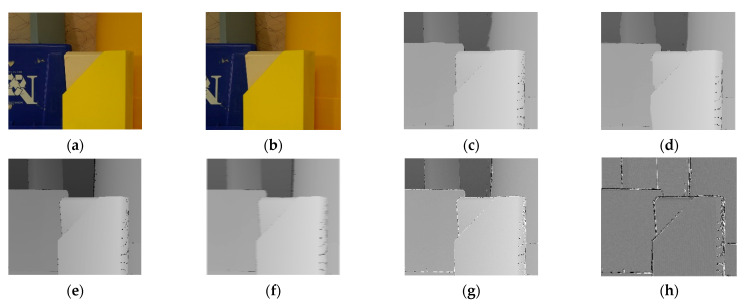
Experimental results for plastic stereo image pair. (**a**) Left view; (**b**) right view; (**c**) Siamese network algorithm; (**d**) generative adversarial network algorithm; (**e**) region matching algorithm; (**f**) DP algorithm; (**g**) SIFT matching algorithm; (**h**) proposed algorithm.

**Figure 9 sensors-24-01807-f009:**
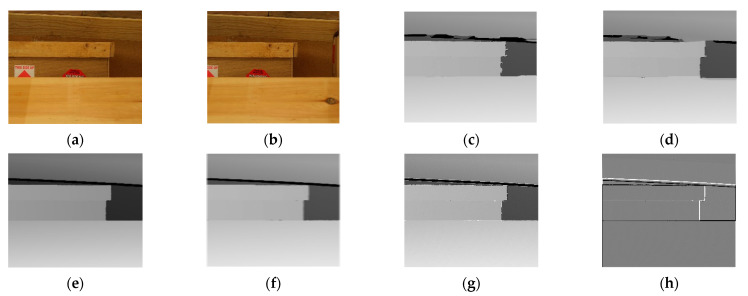
Experimental results for wood stereo image pair. (**a**) Left view; (**b**) right view; (**c**) Siamese network algorithm; (**d**) generative adversarial network algorithm; (**e**) region matching algorithm; (**f**) DP algorithm; (**g**) SIFT matching algorithm; (**h**) proposed algorithm.

**Table 1 sensors-24-01807-t001:** Image matching results.

Algorithm Name	False Match Rate	PSNR
Siamese Network for Feature	14.75%	30 dB
GAN-based Feature Matching	13.47%	30 dB
Region Matching Element-Based Feature Matching	7.49%	33 dB
Dynamic Programming	8.59%	27 dB
SIFT-Based Feature Matching	5.42%	32 dB
Proposed Algorithm	2.02%	34 dB

**Table 2 sensors-24-01807-t002:** Matching results of feature points with varied image brightness.

Objective Index	10 cd/m^2^	20 cd/m^2^	30 cd/m^2^	40 cd/m^2^	50 cd/m^2^
**Correct rate**	97.61%	97.63%	97.57%	97.59%	97.62%

**Table 3 sensors-24-01807-t003:** Measurement results data.

Group Number	Measured Length/mm	True Length/mm	Relative Error/%	Cost Time/s
1	53.21	54.26	1.93	0.74
2	57.89	58.16	0.46	0.68
3	73.38	73.29	0.12	0.59
4	76.75	78.01	1.61	1.12
5	81.39	82.67	1.54	0.88
6	84.57	84.69	0.14	0.79
7	92.59	92.12	0.51	0.76
8	94.57	95.31	0.77	0.97
9	105.34	104.99	0.33	0.89
10	107.89	108.01	0.11	0.89

## Data Availability

Publicly available datasets were analyzed in this study. This data can be found here: https://github.com/xzyxzy1/Measurement (accessed on 1 February 2024).

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
