# Peer review of "Binocular Visual Measurement Method Based on Feature Matching"

_sensors, 2024, doi:10.3390/s24061807_

Round 1
Reviewer 1 Report (New Reviewer)
Comments and Suggestions for Authors
The proposed algorithm significantly improves the matching accuracy while reducing the complexity of the algorithm.
Author Response
I would like to express my sincere gratitude for your insightful feedback and valuable comments on my manuscript. Your constructive criticism and thorough evaluation have immensely contributed to the improvement of the quality and clarity of the paper. Your expertise and attention to detail have been instrumental in guiding me towards addressing key areas for enhancement. Your dedication to ensuring the rigor and integrity of scientific research is truly commendable, and I am grateful for the time and effort you have invested in reviewing my work.
Thank you once again for your invaluable contribution to the advancement of scientific knowledge in our field.
Reviewer 2 Report (New Reviewer)
Comments and Suggestions for Authors
In this study, three-dimensional measurement using image processing based on a binocular vision sensor, several important measurements and parameters play a crucial role such as: disparity map, depth map, baseline length, camera calibration parameters, image resolution, accuracy and precision, processing time, environmental factors, calibration stability, and stereoscopic vision quality. Understanding and optimizing these measurements contribute to the overall effectiveness of three-dimensional measurement systems based on binocular vision sensors.
Some comments
1. Firstly, what is the author’s opinion for the above?
2. What are the specific issues addressed in the abstract related to the use of binocular cameras for object detection?
3. How does the proposed method aim to improve three-dimensional measurement accuracy?
4. What is the key performance metrics mentioned in the abstract for evaluating the proposed algorithm?
5. How does the proposed algorithm demonstrate robustness against certain challenges mentioned in the abstract?
6. What are the main improvements highlighted in the conclusion resulting from the proposed method?
7. Which algorithms were compared in the experiments, and what were the criteria for the comparison?
8. What datasets and testing platform were utilized for the experiments mentioned in the conclusion?
9. How well does the proposed algorithm meet the expected requirements, and what are those requirements?
The introduction provides a good overview of the general context and challenges associated with three-dimensional measurement using binocular vision sensors. It mentions the advantages of binocular vision, its applications in various fields, and the fundamental principle based on the parallax principle. The challenges related to the fast and accurate matching of homonymous pixels are highlighted, and different existing matching methods, both deep learning-based and artificial feature-based, are briefly discussed.
The introduction also introduces the proposed method, emphasizing the combination of feature points and feature regions, the use of FAST for feature extraction, and the employment of the Particle Swarm Optimization algorithm for determining the optimal search range and seed points. It mentions the use of ZNCC as a similarity measure and the triangulation formula for calculating three-dimensional information. The concluding paragraph summarizes the improvements claimed by the proposed algorithm and mentions experimental results demonstrating its effectiveness.
However, to enhance the introduction and provide more context for readers, you may consider the following additions or modifications:
10. Explicitly state the motivation for the research. Why is it essential to improve the accuracy of three-dimensional measurement using binocular vision sensors? Are there specific applications or industries where this improvement is particularly crucial?
11. Provide a brief overview of the current state of the field, emphasizing the existing challenges and limitations in three-dimensional measurement using binocular vision sensors. This can help readers understand the gap that the proposed method aims to fill.
12. Clearly articulate the significance of the proposed method. How does it address the current limitations in the field? What potential impact can it have on applications in robotics, industrial automation, medical imaging, etc.?
13. Highlight the unique contribution of the proposed method compared to existing approaches. What sets it apart, and how does it address the challenges mentioned in the introduction more effectively?
14. Briefly mention the scope of the study and any limitations associated with the proposed method. This provides transparency and helps manage reader expectations.
15. Adding these elements can provide a more comprehensive introduction, giving readers a clearer understanding of the research context, motivation, and the unique contributions of the proposed three-dimensional measurement method based on binocular vision sensors.
Author Response
Please see the attachment.

Reviewer 3 Report (New Reviewer)
Comments and Suggestions for Authors
The work is about using binocular vision to measure a scene in three dimensions. The authors selected some feature matching techniques and proposed combining them to improve the final result using known heuristics. Technically, the work provides interesting results and adds some value to the state-of-the-art. It is not an extremely new work, and it is not difficult to find other works along the same lines, with the same ideas and using other feature matching and information fusion techniques. The text is generally good, but some of the figures need attention: the captions for the figures with images of the experiments only describe part of the figures and not all of them. The references seem adequate, considering that this is an area that has already been extensively explored. The title should be reformulated, because it is too generic for this paper.
Comments on the Quality of English LanguageThere are few typos and no strong errors were found.
Author Response
Please see the attachment.

This manuscript is a resubmission of an earlier submission. The following is a list of the peer review reports and author responses from that submission.
Round 1
Reviewer 1 Report
Comments and Suggestions for Authors
The authors propose a particle swarm optimization to improve binocular machine vision. The idea and the proposed method is interesting. Unfortunately, the draft is very hard to read. It does not comply to the standards of academic writing. I would suggest that the authors rewrite their draft and submit it again, if the many disctracting small mistakes have been removed.
Some of many more examples:
o Several formulas are larger than the allowed text width.
o There is a problem with punctuation. A lot of spaces are missing throughout the draft. This makes the draft very hard to read.
o Line 146 'angle' instead of 'Angle'
o The meaning of the sentence 'The idea of poles and poles...' in line 112 is not clear to me
Comments on the Quality of English Languagesee above
Reviewer 2 Report
Comments and Suggestions for Authors
Th authors propose a method that claims to improve stereo matching in terms of both computation time and accuracy. The core piece of novelty claimed seems to be a particle swarm optimization based region growing algorithm.
There are several serious issue with the paper in its current form and hence I do not recommend the publication of the paper in its current form.
The paper is not well written and requires substantial editing to make things clear to the readers. A good example would be Table 1. To begin with the table title does not make any sense. It looks like the authors forgot to change the default title, that perhaps came from the template. More importantly, there is no way to identify the references for the works being used for comparison. To the best of my understanding the papers being compared to have not been even cited. Such sloppy work required to be pointed out.
There are many methods out there (like the one cited below) which has proved to be fast and accurate on many standard stereo vison benchmarks. The authors skip comparisons to any of these methods. More importantly, the authors do not even make an effort to describe their dataset well nor do they provide any comparisons of their method on standard benchmark datasets.
Tankovich, Vladimir, et al. "Hitnet: Hierarchical iterative tile refinement network for real-time stereo matching." Proceedings of the IEEE/CVF Conference on Computer Vision and Pattern Recognition. 2021.
The results section is also very poorly written without any proper description of the tables and plots.
To summarize, there are three important changes required for the paper to be considered for publication
1. The authors should do a much better job in reviewing the state of the art and include some of the latest work in that section, commenting on why their method is better than those prior work
2. They should compare their work on a few standard stereo vison benchmarks to test how well the results generalize and also to verify its robustness.
3. They should also do a substantially better job in writing the paper well. For example the authors are expected to describe and discuss the results presented in tables and plots in detail rather than just mentioning that the results can be seen in a table without actually commenting on it.
Comments on the Quality of English LanguageThere are multiple language issues starting from very basic issues like adding a space after commas and full stops to the issues with the grammar.
As an example, consider the following sentence from the paper.
"In order to solve the problem that the particle swarm optimization algorithm is easy to converge to the local minimum,an improved particle swarm optimization algorithm is designed in this paper"
This is a rather convoluted way of saying that the PSO algorithm has a tendency to get stuck in local minimums and the paper is proposing a solution to that problem.
Reviewer 3 Report
Comments and Suggestions for Authors
See attached docx file for suggestions/comments/corrections

See attached docx file for suggestions/comments/corrections